# An Assessment of MT1A (rs11076161), MT2A (rs28366003) and MT1L (rs10636) Gene Polymorphisms and MT2 Concentration in Women with Endometrial Pathologies

**DOI:** 10.3390/genes14030773

**Published:** 2023-03-22

**Authors:** Kaja Michalczyk, Patrycja Kapczuk, Grzegorz Witczak, Piotr Tousty, Mateusz Bosiacki, Mateusz Kurzawski, Dariusz Chlubek, Aneta Cymbaluk-Płoska

**Affiliations:** 1Department of Gynecological Surgery and Gynecological Oncology of Adults and Adolescents, Pomeranian Medical University in Szczecin, 70-111 Szczecin, Poland; 2Department of Biochemistry and Medical Chemistry, Pomeranian Medical University, Powstańców Wlkp. 72, 70-111 Szczecin, Poland; 3Department of Obstetrics and Gynecology, Pomeranian Medical University, Powstańców Wielkopolskich 72, 70-111 Szczecin, Poland; 4Department of Functional Diagnostics and Physical Medicine, Pomeranian Medical University in Szczecin, 70-111 Szczecin, Poland; 5Department of Experimental and Clinical Pharmacology, Pomeranian Medical University in Szczecin, Powstancow Wielkopolskich 72, 70-111 Szczecin, Poland; 6Department of Reconstructive Surgery and Gynecological Oncology, Pomeranian Medical University in Szczecin, Al. Powstańców Wielkopolskich 72, 70-111 Szczecin, Poland

**Keywords:** endometrial cancer, uterine cancer, gene polymorphism, MT1A, MT2A, MT1L

## Abstract

Several studies have indicated a relationship between metallothionein (MT) polymorphisms and the development of different pathologies, including neoplastic diseases. However, no studies thus far have been conducted on the influence of MT polymorphisms and the development of endometrial lesions, including endometrial cancer. This study included 140 patients with normal endometrial tissue, endometrial polyps, uterine myomas and endometrial cancer. The tissue MT2 concentration was determined using the ELISA method. MT1A, MT2A and MT1L polymorphisms were analyzed using TaqMan real-time PCR genotyping assays. We found no statistical difference between the tissue MT2 concentration in patients with EC vs. benign endometrium (*p* = 0.579). However, tissue MT2 concentration was significantly different between uterine fibromas and normal endometrial tissue samples (*p* = 0.019). Menopause status did not influence the tissue MT2 concentration (*p* = 0.282). There were no significant associations between the prevalence of MT1A, MT2A and MT1L polymorphisms and MT2 concentration. The age, menopausal status, and diabetes status of patients were identified as EC risk factors.

## 1. Introduction

Endometrial cancer (EC) is the most common gynecological malignancy, yet its incidence is still rising with little improvement in patient survival [1]. It is the sixth most common cancer in women worldwide. In 2020, in accordance with the World Cancer Research Fund International, the global incidence of EC was 417,367 patients, with an ASR (age-standardized rate) of 8.7 per 100,000. Poland was found to have the highest rate of endometrial cancer diagnosis at 26.2 ASR/100,000 [2]. The International Agency for Research on Cancer estimates that the incidence rate of endometrial cancer will rise by more than 50% worldwide by 2040 [3]. It is a heterogeneous disease that varies in molecular characteristics and patient prognosis. Despite numerous studies investigating the pathophysiology and mechanisms responsible for the formation of endometrial cancer, the pathogenesis is still not fully understood. The formation of endometrial cancer involves multiple molecular pathways and epigenetic changes. Despite the red flag symptoms, including vaginal bleeding, that often allow for the early detection of an endometrial carcinoma, there are still no highly specific or sensitive biomarkers. The main risk factors for endometrial cancers include elevated estrogen levels and obesity, especially in postmenopausal women [4]. On the other hand, the use of combined estrogen–progestin hormonal therapy and physical activity were found to be protective of EC [5,6].

Metallothioneins (MTs) are a group of cysteine-rich proteins (MT-1 through MT-4) located on chromosome 16q13. They are major intracellular zinc-binding proteins responsible for zinc uptake, distribution and storage. They also have a regulatory role in the transportation, protection against oxidative stress and toxic effects of trace metals and other particles, including Copper, Cadmium, Lead and glutathione [7,8]. They also participate in processes that include the metabolism of free radicals and apoptosis [7]. In physiological conditions, they are usually expressed at low levels [9]. Their synthesis was found to increase during oxidative stress to protect cells against potential cytotoxicity, radiation or DNA damage [10,11,12,13]. They are involved in the pathophysiology of various diseases, including cancer [14,15], yet their role remains unknown. As metallothioneins were found to have antiapoptotic, antioxidant, proliferative and angiogenic effects, there is an increased focus on determining their role in oncogenesis, tumor progression, response to cancer therapy and patient prognosis [16]. Multiple studies have demonstrated an increased expression of both tissue and serum MT levels in lung, kidney, prostate, testes, urinary bladder, pancreatic, cervical and endometrial cancers. In some, the expression of MT was found to correlate with the tumor staging, grading, treatment resistance and prognosis [14,17,18]. However, the expression of MT seems to differ between the tumor types and may depend on the type of tumor differentiation status, its environment or associated gene mutations [14].

The prevalence of multiple gene polymorphisms was found to be associated with an increased cancer risk [19,20]. For our study, we selected gene polymorphisms characterized by a relatively high frequency to assess their influence on endometrial cancer risk in association with MT2 concentration. The polymorphisms of MT1 and MT2 were previously reported to be directly involved in malignant transformation, xenobiotic metabolism and/or oxidative stress processes. In our study, we decided to determine the presence of MT1A (rs11076161), MT2A (rs28366003) and MT1L (rs10636) polymorphisms among the studied patients as research on the influence of the selected polymorphisms on the risk of developing endometrial cancer has not yet been conducted. This study aimed to determine the potential of MTs as biomarkers for endometrial cancer diagnosis, with regard to other risk factors such as obesity and diabetes. The knowledge about MTs may provide a new insight into endometrial cancer diagnostics, especially in correlation with other clinical examinations.

## 2. Materials and Methods

This study consisted of 140 patients treated at the Department of Gynecological Surgery and Gynecological Oncology of Adults and Adolescents, Pomeranian Medical University. It was a case—control study of patients with a confirmed diagnosis of endometrial cancer based on histopathological evaluation who were admitted for a radical surgery at the time of the study. The control group consisted of patients admitted for hysteroscopy or myomectomy. Patients suffering from chronic diseases, the recurrence of endometrial cancer, previous cancer treatment or other types of primary cancer were excluded from the study. Patients with incomplete or missing data were also removed from the study group. Finally, 110 patients were included in the final analysis. Informed consent to participate in the study was obtained from both patients and controls. Patient characteristics are demonstrated in Table 1. Of the included 21 patients diagnosed with endometrial cancer, 19 were diagnosed with endometrioid carcinoma, 1 with a serous EC and 1 with an undifferentiated carcinoma. For tumor characteristics, the patients’ FIGO (The International Federation of Gynecology and Obstetrics) staging, tumor grading and the presence of lymph node metastases were assessed. The research was conducted in accordance with the Helsinki Declaration and with the consent of the Ethics Committee of Pomeranian Medical University in Szczecin, under the number KB-0012/27/2020, on 9 March 2020.

### 2.1. Blood Sample and Tissue Collection

Blood samples were taken from patients at the time of hospital admission. Two blood samples were collected. One was centrifuged, while the other was directly frozen. The tissue specimen was collected during the surgery, either during a hysteroscopy/laparoscopy or laparotomy. The specimens were stored at a temperature of −80 degrees Celsius.

### 2.2. Measurement of Tissue MT2

The tissue MT2 concentration was determined using the ELISA method. The collected fragments of carcinomas, fibroids, polyps and normal endometrial tissues underwent knife homogenization in the liquid nitrogen. To homogenize, a PBS–potassium phosphate (pH 7.4) lysis buffer was used containing NaCl 0.14M; KCl 0.0027M; Phosphate buffer, pH 7.4; 0.010M (Witko, Poland). Homogenates then underwent a 20 min sonification at 4 °C and were centrifuged at 15,000× *g* for 20 min at 4 °C. Supernatants were stored at −80 °C for later analysis.

The concentration of Human Metallothionein-2 was calculated based on the protein quantity in the specimens. Protein concentrations were measured using a MicroBCAPierce ™ kit (Thermo Fisher Scientific, Waltham, MA, USA), according to the manufacturer’s instructions, and with a plate reader (BiochromAsys UVM 340, Biochrom, Cambridge, UK) at 562 nm. The concentration of Human Metallothionein-2 was measured with the Human Metallothionein-2 (MT-2) ELISA kit (MyBioSource, San Diego, CA, USA, Cat # MBS703385), according to the manufacturer’s instructions.

The extracted material was defrosted, and the reaction mixtures were transferred onto an ELISA microplate, following the manufacturer’s protocol. The optimum dilutions were selected by checkerboard titration. Sandwich ELISAs were performed using 96 well microtiter plates coated with MT2 biotin-conjugated monoclonal antibodies. Finally, horseradish peroxidase (HRP) and a 3,3’,5,5’-Tetramethylbenzidine (TMB) substrate were added to the wells. There was a visible color change. After stopping the reaction with the Stop Solution, the absorbance was read at 450 nm using Biochromasys UVM 340 technology. The signal intensity was directly proportional to the amount of MT2 in the sample. Concentrations were expressed in ng/mg of protein.

### 2.3. Single-Nucleotide Polymorphism Analysis

The patients were genotyped for the following single-nucleotide polymorphisms (SNPs) within MT1A, MT2A and MT1L. The following variants were genotyped: MT1A rs11076161 A > G, MT2A rs28366003 A > G and MT1L rs10636 G > C. Genomic DNA was isolated from 0.2 mL of whole blood samples, using a commercial kit for genomic DNA isolation and the Genomic Mini AX Blood 1000 Spin (A&A Biotechnology, Gdańsk, Poland). TaqMan real-time PCR genotyping assays (Thermo Fisher Scientific) were used for the detection of the studied SNPs (Assay IDs: C_1402094_10, C_60284591_10, C_25996927_10).

### 2.4. Statistical Analysis

The statistical analysis was performed using Statistica 10 software. The results were presented as the mean ± SD or absolute frequencies and percentage values, in accordance with the type of the variable. The unpaired, two-sided Student’s *t*-test, the chi-square test and the Mann-Whitney test were used to compare the sociodemographic, clinical and questioned data between the groups. Significance was defined at *p* < 0.05. The Mann–Whitney U test was used to compare the studied group with the control group. Kruskal–Wallis and post hoc tests were used to measure the differences between multiple groups. The Spearman correlation was created to check for any correlations between the measured parameters. The occurrence of SNPs was estimated by an odds ratio (OR) analysis with a 95% confidence interval, using univariable logistic regression models.

## 3. Results

### 3.1. Tissue MT2 Concentrations

In our study, there was no statistical difference between the tissue MT2 concentration in patients with malignant vs. benign endometria (*p* = 0.579). As a part of the study, we also evaluated for any differences between other groups, i.e., cancer vs. uterine fibroma, cancer vs. normal endometrial tissue and cancer vs. endometrial polyp. There was only a significantly important difference in tissue MT2 between the groups of patients diagnosed with uterine fibromas and presenting with normal endometrial tissue (*p* = 0.019). We found no difference in the tissue MT2 concentration between the patients before and after menopause (*p* = 0.282). Patients’ BMI was found to influence the tissue MT2 concentration, with higher MT2 concentrations found in patients with a greater body weight (r = 0.01, *p* < 0.05).

### 3.2. Associations between SNPs and Tissue MT2 Concentration

We found no associations between prevalence of the selected polymorphisms and MT2 concentration. The most frequent genotype of MT1A among the studied population was GG, with only four patients with the AA genotype. Regarding MT2A, none of the patients were found to have the GG genotype. Specific results are listed in Table 2.

### 3.3. Regression Analysis

We performed a univariate logistic regression analysis to assess whether any of the selected genotypes influenced the risk of endometrial cancer. The results of the study showed no influence of MT polymorphism on the endometrial cancer risk (Table 3).

As part of the study, we attempted to assess the risk factors for endometrial cancer. The patients’ age (OR 7.26, *p* = 0.001), menopausal status (OR 14.77, *p* = 0.001), and diabetes status (OR 16.60, *p* < 0.0001) were found to increase the risk of endometrial cancer. However, we did not find any associations between MT2 concentration and EC risk. Specific results are listed in Table 4.

## 4. Discussion

For many years, the division of endometrial cancer subtypes was based on the Bokhman classification, dividing the groups of patients into estrogen dependent (type I) and estrogen independent (type II). The Cancer Genome Atlas (TCGA) project has identified a new molecular classification of endometrial cancer, with four distinct prognostic endometrial cancer subtypes based on genomic abnormalities. The classification differentiates POLE mutated, mismatch repair-deficient, p53 abnormal and no specific molecular profile groups of patients. In accordance with the ESGO/ESTRO/ESP guidelines, patients should be classified based on their molecular profiles and FIGO staging to define prognostic risk groups and to decide on the need of additional/adjuvant treatment [21]. With the rising prevalence of endometrial cancer [22], we not only need better prognostic markers but also diagnostic factors that allow for early cancer detection. Some potential biomarkers for endometrial cancer have been identified, including the most common somatic mutations in already well-known tumor-suppressor genes and oncogenes, such as the PTEN, TP53, POLE and KRAS mutations. The utility of screening for endometrial cancer is limited and should only be considered for high-risk populations, e.g., patients with type 2 Lynch Syndrome [23]. Apart for the use of serum biomarkers, the use of a transvaginal ultrasound is possible as it is a reasonably sensitive and specific screening method [24]. As the sensitivity and specificity of the already-known EC diagnostic markers is limited, we decided to study the potential role of metallothioneins in endometrial cancer diagnosis.

Metallothioneins were found to participate in carcinogenesis and to have important roles in tumor growth, progression, metastasis and drug resistance. Their effect can be caused by their role in the regulation of cell cycle, proliferation and apoptosis; their participation in multiple cell signaling pathways, zinc homeostasis, tumor cell microenvironment remodeling and cell adhesion and migration; their role in angiogenesis pathways, including VEGF and MMP-9, and their role in the mediation of p53 and NF-kB activity [14]. They are intracellular proteins that are able to form complexes with group IIB metals, thus being able to participate in heavy metal detoxification [25]. Their gene transcription is activated by stress stimuli including metals, glucocorticoids, catecholamines, ROS (reactive oxygen species) and proinflammatory cytokines [26,27]. In humans, four MT isoforms have been identified (MT1, MT2, MT3 and MT4), and they are all located on chromosome 16q13 with functional genes for their isoforms [26,28].

Previous studies have shown the overexpression of MT to be a prognostic marker for tumor progression and drug resistance in, i.a., ovarian [25,29], breast [30], lung [31], renal [32], bladder [33] and oral cancer [34] and melanoma [35]. Yet in some tumors, MT levels are downregulated (i.a., hepatocellular, gastric, colorectal and central nervous system). Due to the involvement of MT in cell proliferation, increased concentrations of MT in cancer patients were discovered to be associated with a poor patient prognosis [28]. Large discrepancies between MT expression have been found between different tumor types, and yet there is no reliable explanation for the associations between MT expression in tumor tissue, patient prognosis nor resistance to treatment [16]. Gumulec et al. conducted a meta-analysis to summarize the evidence of the studies on MT as molecular markers of various types of cancer [15]. The authors analyzed 77 studies that included 8015 tissue samples, revealing a positive correlation between MT expression and head and neck and ovarian tumors. A negative association was found for liver tumors, while no significant associations were found for breast, colorectal, prostate, thyroid, stomach, bladder, kidney, gallbladder and uterine cancer, nor for melanoma.

A study on gynecological malignancies by Ioachim et al. [36] revealed a statistically significant difference in MT expression between endometrial carcinoma and simple hyperplasia tissues. In patients with carcinomas, the expression correlated positively with histopathological grading and inversely with progesterone receptors.

In our study, we decided to evaluate the MT concentration in patients with different uterine diagnoses: patients with normal endometrial tissue, endometrial polyps, uterine myomas and endometrial cancer in patients presenting with abnormal uterine bleeding. Endometrial polyps are caused by the benign hyperplastic growth of endometrial tissue, while uterine fibroids are caused due to an imbalance between the cells’ proliferation rate and their apoptosis [37]. Both of these conditions can be related to hypoestrogenism [37,38]. Endometrial hyperplasia is a condition also associated with increased estrogen; however, it is an estrogen-driven precursor lesion to endometrial endometrioid adenocarcinoma characterized by an increased gland to stroma ratio of >3:1 glandular to stromal elements [39,40,41]. We found no statistically significant differences between MT2 expression in the tissue samples of patients with endometrial cancer and those with benign tissue (*p* = 0.579). We have also evaluated changes in the MT2 concentration between benign lesions. There was a significant difference in MT2 concentration between uterine fibromas and normal endometrial tissue (*p* = 0.019). A study by Klimek et al. [42] evaluated changes in MT levels during different menstrual cycle phases and revealed that MT expression changes respective to hormonal fluctuations, with the highest observed during the mid-secretory phase and its respective decrease occurring during the early, late secretory and mid-proliferative phases, suggesting a possible role of MT in the protection of endometrial cells against apoptosis. A similar study suggested higher MT expression during phases associated with low circulating progesterone levels [43]. As uterine fibroids are hormone-dependent and estrogen is considered the major mitogenic factor in the uterus [44], the expression of MT in fibroid tissue may be related to the rapid proliferation and presence of hormone receptors in the tissue. Additionally, tumor growth, whether benign or malignant, requires the formation of new blood vessels to obtain sufficient oxygen and nutrition for further growth and progression. Studies have demonstrated an important role of MTs in tumor angiogenesis. A study by Miyashita and Sato showed increased MT1 expression in vascular endothelial cells at the site of angiogenesis, and MT1 downregulation was found to cause the inhibition of cell proliferation and angiogenesis [45]. In our study, we did not ask the patients for any use of hormonal therapy, nor we did not evaluate the progesterone/estrogen receptor status of endometrial cancer tissues. Further studies assessing the use of exogenous sources of estrogen may provide additional insight on the correlation between estrogens and MT status.

MTs were also discovered to regulate multiple proteins and transcription factors essential for intracellular signaling pathways, including CuZn-superoxide dismutase, Zn-finger proteins, and proapoptotic proteins (e.g., p53 [17,28,46]). P53 gene mutation resulting in p53 is found in numerous cancers, including endometrial cancer, with a p53-abnormal subgroup of patients with the poorest prognosis among all endometrial cancer patients [47]. Both in vitro and in vivo studies have demonstrated a strong positive correlation between p53 mutation and elevated MT I and MT II concentrations [48]. In our study, due to the limited population of endometrial cancer patients, we did not analyze the p53 mutation status in the endometrial cancer patients. However, further studies on endometrial cancer that assess the correlation between MT concentrations and p53 mutation status should be performed to determine any possible relationships.

SNPs can alter gene expression and influence protein activity; thus, it was suggested that genetic variations of MT isoforms may influence cancer susceptibility [49,50,51,52,53,54]. In humans, MTs have four main isoforms (MT1, MT2, MT3 and MT4) and can be further subdivided into functional subisoforms. Although the expression of MTs is not universal in all tumor types, increasing evidence suggests that the differential expression of particular MT isoforms can be utilized for tumor diagnosis and therapy [14]. As a part of this research, we measured the most common isoforms, including MT1A, MT2A and MT1L. MT2A is a MT2 isoform which was found to be associated with an increased risk of prostate [53], laryngeal [51] and breast cancers [52]. MT2A was found to be the most expressed MT isoform in breast tissue, and its expression was found to be positively correlated with histological grading [17]. Its expression was also found to be associated with an increased recurrence rate and poorer survival in patients suffering from ductal breast carcinomas [16,18]. Moreover, studies on breast cancer revealed an inverse correlation between the expression of MTs and progesterone and estrogen receptors [55]. Similar to breast cancer, endometrial cancer is a hormone-dependent malignancy, and unopposed estrogen is one of the major risk factors [56,57]. We decided to check for similarities. From the pathophysiologic perspective of endometrial cancer, a type 1 endometrioid carcinoma is associated with a prolonged elevation of estrogen levels leading to the persistent stimulation and proliferation of the endometrial tissue. The causes of increased estrogen concentration may be correlated with different EC risk factors and include obesity and the use of estrogen-based hormonal therapy, especially when unopposed by progesterone, tamoxifen, ovarian cortical hyperplasia, polycystic ovarian syndrome and the presence of estrogen-producing tumors [58].

A study by Białkowska et al. [59] evaluated the gene polymorphisms of MT2A, MMP-2, MMP-7, and MMP-13. The authors did not find any significant association between gene polymorphisms, their relation to serum Zn level, and the occurrence of breast, lung or colon cancer. Other studies have shown an association between MT2A-5A/G SNP and an increased risk of breast cancer [50,52], prostate cancer [53,60] and gastric adenocarcinoma [61]. In prostate cancer, Krześlak et al. [60] found a significant association between MT2A SNP and Cd, Zn, Cu and Pb levels, suggesting that SNP polymorphisms may affect MT2A gene expression and be associated with metal accumulation. A study by Nakane et al. [62] evaluated the impact of MT gene polymorphisms on the risk of lung cancer. The authors found MT-1A C/A, MT-1B G/A and MT-1F C/T variants to significantly increase the risk of lung cancer. MT-1A polymorphism was also previously found to be associated with an increased risk of oral cancer [54]. In our study, we found no associations between the prevalence of the selected polymorphisms and MT2 concentration. The prevalence of specific genotypes was not associated with metalloestrogen or MT2 concentration. None of the assessed SNPs were associated with an increased EC risk.

Our study is one of the first to evaluate the influence of metallothionein 2 and MT gene polymorphism on endometrial cancer risk. Nevertheless, there are some limitations to the study. The population study was limited to 110 consecutive patients treated at the clinic. Initially, 140 patients were admitted to the population sample. However, due to incomplete or missing data, 30 patients were excluded from the final study group. In our study, we decided to evaluate changes in trace metal levels, metallothioneins and their polymorphisms between not only endometrial cancer and normal endometrial tissue but also in different benign uterine pathologies. Due to the limited population of EC patients, we did not divide them based on histopathological or molecular characteristics. Different suppressor genes and oncogenes have been found among different histopathological subgroups of patients (i.e., the PTEN mutation associated with endometrioid endometrial carcinoma and the p53 mutation in serous endometrial carcinoma [63]). Further studies on greater populations are needed to confirm our findings and to determine if there are any differences in MT levels and their polymorphisms between the different subgroups of EC patients. It would be interesting to conduct further studies that would assess both serum and tissue levels of MTs simultaneously to check for any associations and eventual changes in their expression.

## 5. Conclusions

Having conducted a literature review, we found no previous studies on the relationship between the polymorphisms of MT1A (rs11076161), MT2A (rs28366003) and MT1L (rs10636) and the possibility of developing endometrial cancer. Our analysis demonstrated no correlation between the selected polymorphisms and the risk of endometrial cancer. Moreover, we did not find any influence of tissue MT2 concentration. Menopause status does not seem to influence tissue MT2. The patients’ BMI was found to correlate with increased tissue MT2 expression.

## Figures and Tables

**Table 1 genes-14-00773-t001:** Patient characteristics.

Characteristics	Number of Patients
**Age**	
<50	47
>50–60	35
>60	29
**Cancer**	21
**Staging**	
Figo 1–2	19
Figo 3–4	2
**Grading**	
1	1
2	16
3	4
**Non-cancer**	89
Endometrial polyp	48
Uterine fibroma	25
Normal endometrium tissue	16
**BMI**	
<25	40
>25	70
**Cigarette smoking**	
Yes	7
No	103
**Menopause**	
Yes	66
No	44
**Lymph node metastasis**	
Yes	1
No	20
**Type 2 Diabetes**	
Yes	15
No	95
**Hypothyroidism**	
Yes	18
No	92

**Table 2 genes-14-00773-t002:** Associations between specific MT polymorphisms and MT2 concentration.

Polymorphism	Genotype	Studied Population (*n*)	MT2		*p*-Value
			Below Median (*n*)	Above Median (*n*)	
MT1A	GG	37	0.194	0.227	-
	AG	31	0.237	0.239	0.941
	AA	4	0.174	0.197	0.913
MT2A	AA	66	0.200	0.235	-
	AG	6	0.362	0.195	0.730
	GG	0	-	-	-
MT1L	GG	37	0.213	0.255	-
	CC	5	-	0.180	-
	GC	30	0.201	0.215	0.950

**Table 3 genes-14-00773-t003:** Univariate logistic regression model.

Polymorphism	Genotype	*n* of Patients	Endometrial Cancer	OR	Lower 95% CI	Upper 95% CI	*p*-Value
MT1A	GG	50	11	1	-	-	-
	AG	53	8	0.630303	0.230343	1.724742	0.362382
	AA	7	1	0.590909	0.064161	5.442173	0.453685
MT2A	AA	102	19	1	-	-	-
	AG	8	1	0.624060	0.072417	5.377889	0.665234
	GG	0	0	-	-	-	-
MT1L	GG	59	12	1	-	-	-
	CC	7	0	-	-	-	0.187125
	GC	44	8	0.870370	0.321998	2.352640	0.667486

**Table 4 genes-14-00773-t004:** Influence of the assessed variables on endometrial cancer risk.

		Endometrial Cancer	Control Group	OR	Lower 95% CI	Upper 95% CI	*p*-Value
Age	Above vs. Below Median	17	39	7.26	1.99	26.57	0.001
Menopause	Yes vs. No	19	45	14.77	1.89	115.81	0.001
Grading	2–3 vs. 1	17	2	-	-	-	0.733
Staging	3–4 vs. 1–2	1	0	-	-	-	0.773
Smoking	Yes vs. No	3	4	3.70	0.76	18.08	0.086
Diabetes	Yes vs. No	10	5	16.60	4.72	58.41	<0.0001
MT2 level	Above vs. Below Median	9	27	0.75	0.47	3.72	0.599

## Data Availability

The data presented in this study are available upon request from the corresponding author. The data are not publicly available due to ethical restrictions.

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
