# Peer review of "An Assessment of MT1A (rs11076161), MT2A (rs28366003) and MT1L (rs10636) Gene Polymorphisms and MT2 Concentration in Women with Endometrial Pathologies"

_genes, 2023, doi:10.3390/genes14030773_

Round 1

Reviewer 1 Report

MT1A, MT2A, MT1L & MT2 in endometrial pathology study needs major revision.

1.     Manuscript correction is needed before detailed revision: Discussion line 1-4 hard to understand “Authors Molecular changes in endometrial cancer are usually only linked to the determination of most common somatic mutations in already well-known tumor-suppressor genes and oncogenes including i.e. PTEN, TP53, POLE and KRAS mutations yet also other mutations or gene polymorphisms may have a role in endometrial cancer and may be helpful in endometrial cancer screening”, Conclusion line 1:” .. presious studies” Reference 4 is same with Reference 8 : The Roles of Metallothioneins in Carcinogenesis. J. Hematol. Oncol.

2.     Endometrial pathology: normal endometrium, endometrial hyperplasia and endometrial cancer is generally accepted carcinogenesis pathway .It’s not desirable to add polyp and uterine fibroma in study population. Is there only one endometrial cancer pathology? How about serous, clear cell besides endometrioid type?

3.     Adding picture, figure of MT1A, MT2A, MT1L & MT2 can help understand better for readers and precision medicine. 

Author Response

Dear reviewer,

We would like to thank you for your comments. Please see our answers and the edited version of the manuscript

Discussion line 1-4 hard to understand “Authors Molecular changes in endometrial cancer are usually only linked to the determination of most common somatic mutations in already well-known tumor-suppressor genes and oncogenes including i.e. PTEN, TP53, POLE, and KRAS mutations yet also other mutations or gene polymorphisms may have a role in endometrial cancer and may be helpful in endometrial cancer screening”, -

we have improved the paragraph, please see lines 180-195

Conclusion line 1:” .. presious studies” – sorry for the mistake, we have corrected this

Reference 4 is same with Reference 8 : The Roles of Metallothioneins in Carcinogenesis. J. Hematol. Oncol. – we have asked the editorial office to delete the copied reference

  1. Endometrial pathology: normal endometrium, endometrial hyperplasia and endometrial cancer is generally accepted carcinogenesis pathway. It’s not desirable to add polyp and uterine fibroma in study population. Is there only one endometrial cancer pathology? How about serous, clear cell besides endometrioid type? 

Our study included 21 patients diagnosed with endometrial cancer, of whom 19 were diagnosed with endometrioid carcinoma, 1 with serous EC and 1 with undifferentiated carcinoma - we have added this information to the introduction section

We have also added some rationale as to why we decided to evaluate the different pathologies, please see lines 227-236 and 261-268

  1. Adding picture, figure of MT1A, MT2A, MT1L & MT2 can help understand better for readers and precision medicine. 

We were not sure what the figure should demonstrate; as it is difficult to demonstrate any possible correlations between MTs and their different polymorphism due to limited evidence, we decided not to include any figures

Reviewer 2 Report

In this paper, the authors present the possible implication of metallothionein polymorphisms in the occurrence of endometrial lesions, including endometrial cancer. The subject is of great interest and can be seen as a lead to a new approach in the diagnosis of endometrial neoplasia.
The introduction focuses on the general knowledge on the subject. The information provided is useful in understanding the topic and the importance of the subject in nowadays medicine.
The methods used are complex and reliable.
The inclusion and exclusion criteria were wisely chosen.
The authors conducted a proper collection of the data. The information presented is up to date, suitable, and substantial.
The conclusions are coherent and sustain the findings.
The figures and tables presented are easy to interpret and understand.
Good English level.

Author Response

Dear reviewer, 

We would like to thank you for your comments

We have made some changes to the manuscript (elongated the discussion and added some additional rationale) as suggested by the other editor

Please see the improved version of the manuscript

Round 2

Reviewer 1 Report

.